# Recent Applications of Retro-Inverso Peptides

**DOI:** 10.3390/ijms22168677

**Published:** 2021-08-12

**Authors:** Nunzianna Doti, Mario Mardirossian, Annamaria Sandomenico, Menotti Ruvo, Andrea Caporale

**Affiliations:** 1Institute of Biostructures and Bioimaging (IBB), National Research Council (CNR), 80134 Napoli, Italy; nunzianna.doti@cnr.it (N.D.); annamaria.sandomenico@cnr.it (A.S.); 2Department of Medicine, Surgery and Health Sciences, University of Trieste, 34149 Trieste, Italy; mmardirossian@units.it; 3Institute of Crystallography (IC), National Research Council (CNR), 34149 Trieste, Italy

**Keywords:** retro-inverso peptides, anticancer peptides, drug delivery, peptide antigens, Aβ, IAPP, antimicrobial peptides

## Abstract

Natural and *de novo* designed peptides are gaining an ever-growing interest as drugs against several diseases. Their use is however limited by the intrinsic low bioavailability and poor stability. To overcome these issues retro-inverso analogues have been investigated for decades as more stable surrogates of peptides composed of natural amino acids. Retro-inverso peptides possess reversed sequences and chirality compared to the parent molecules maintaining at the same time an identical array of side chains and in some cases similar structure. The inverted chirality renders them less prone to degradation by endogenous proteases conferring enhanced half-lives and an increased potential as new drugs. However, given their general incapability to adopt the 3D structure of the parent peptides their application should be careful evaluated and investigated case by case. Here, we review the application of retro-inverso peptides in anticancer therapies, in immunology, in neurodegenerative diseases, and as antimicrobials, analyzing pros and cons of this interesting subclass of molecules.

## 1. Topology, Structural Characteristics

As therapeutic agents, peptides have fascinating properties, such as very high specificity and binding affinity, generally low toxicity, and low risk of drug interactions [1]. Moreover, due to the high diversity which is sequence- and structure-dependent, they can be designed as potential drugs to target almost any disease. On the other hand, natural peptides, due to their low size and high sensitivity to most proteases, are quickly excreted or anyhow degraded, resulting in poor biodistribution, bioavailability, and rapid clearance [2], and thus limited therapeutic potential [2]. To increase peptide half-life, stability and bioavailability, many approaches have been proposed including PEGylation, backbone modifications, cyclization, side chain stapling, and lipidation [3,4]. Among these, modification of the backbone is one of the most invasive approaches, as it may profoundly affect the conformation of peptides, especially when it involves alteration of the residue’s isomerization.

All amino acids (except glycine) possess chiral centers and occur in nature almost exclusively as L-enantiomers in proteins and natural peptides. D-amino acid-containing peptides are also found in nature, mainly in some frog species and bacteria, as a result of post-translational modifications [5].

Compared to L-amino acid peptides, D-peptides exhibit an innate resistance to enzymatic degradation and as such have acquired a special importance as potential biopharmaceuticals [6,7,8]. However, given the strong correlation between the structural properties of single residues and of the peptide molecule they belong to, partial or complete modification of peptide isomerism leads in most cases to reduction or even suppression of biological activity [9]. An elegant and often successful solution to translate biologically active L-peptides into D-analogues, is the use of retro-inverso (RI) peptides, which incorporate D-amino acids as stable surrogates of L-amino acids [10], but presented in a reverse (retro) order compared to the parent molecule [8]. RI analogues of all L-peptides, also known as retro-all-D or retro-enantio peptides, are thus peptides composed of D-amino acids introduced in the sequence in reverse direction. The importance of this subclass of peptides is that, when viewed in a fully extended conformation, the side-chains are superimposable with those of the parent L-peptide but with inverted amide bonds and N/C terminal groups [11]. Therefore, in those cases where activity is mostly associated to the array of side chains without any significant contribution from the backbone chemical groups and from the tridimensional organization, a RI analogue has the potential to achieve the same functions as the all-L parent peptide, but with superior stability toward proteolytic degradation [10]. In this regard, retro-inversion is commonly efficacious when one starts from unstructured peptides [12] which work by inductive adaptation on the interacting surfaces and whose array of side chains are more likely to adopt a topology similar to that of the parent peptide. This occurs most readily when the enthalpic contribution of the interactions established by the CO and NH groups of the backbone to stabilize preferential conformations and to bind the target, is relatively poor and can easily be overcome by rearrangements of the retro-inverso analogues which then gain access to the same conformational space of the parent peptide [13]. On these bases, this kind of quasi-neutral isomerization sometimes succeeds and sometimes fails [14,15,16], but in either case provides an interesting and valuable approach to understand the importance of structural elements in molecular recognition events. In particular, when the introduction of D-amino acids with reversal of the peptide backbone and of the C/N terminal groups leave unchanged the peptide binding properties and/or activities, clearly the role of the peptide backbone to the interaction is negligible. This observation was evidently demonstrated by Ruvo et al. [17] studying a very small bioactive peptide. They found that a close topochemical relationship existed between the parent tripeptide MYF-NH2 (three letters amino acids code: L-Met-L-Tyr-L-Phe), and its corresponding retro-inverso isomer (D-Phe-D-Tyr-D-Met) (Figure 1). In the two peptides, the amide bonds of the backbone were interchanged, whereas the 3D orientation of the side chains and the position of the amino group were identical. They demonstrated that the RI isomer retained the binding properties for the protein ligand, the receptor neurophysin II (NP II), and an affinity similar to that of the parent peptide [17]. In this case-study, the successful conversion toward the RI analogue was made possible due to the small size of the molecule analyzed in addition to regeneration of the amino group through a reduction reaction. In larger and structurally more complex peptides, the limitations described above must be taken into account in order to achieve equally effective translations. Indeed, the conversion from an all-L-parent peptide to its retro-inverso analogue implies that the values of the original φ and ψ dihedral angles are exchanged (retro conversion) and transformed to the identical negative values (chiral inversion) in all residues (Figure 1) [16,18,19]. As shown, an identical relationship holds between the inverso and the retro analogues which can be considered as the retro-inverso the one of the other. The angles of the parent peptide become the angles of the retro-peptide and *vice versa* and the same is true with the inverso and the retro analogues (Figure 1). As result, in the retro-inverso analogues, the direction of the peptide bonds is reversed while the side-chain orientation of the amino acid residues is retained. Since the bond lengths of the CO and NH groups are comparable, the positions of the side chains do not change significantly [13] and the two molecules appear as almost identical. Assuming that the activity of a peptide depends mainly on the interactions that the side chains establish with the surface of the target, the peptide functions can be therefore theoretically preserved [20]. Though, since recognition is also often mediated by backbone interactions and is governed by the molecule 3D organization, RI analogues are likely to successfully mimic the precursor molecule only in a restricted number of cases. 

The hydrogen bonds between the CO acceptors and NH donors generate a network of highly stabilizing interactions in peptides arranged as α-helices and β-sheets. If the network of stabilizing interactions is removed, the stability of the 3D structure will be severely compromised in the retro-inverso mimetics, largely affecting their activity [13]. From a topological point of view, the RI analogues of larger peptides [21,22] could adopt conformations similar to that of the parent peptide when the full-length protein or part of it predominantly contains structural elements whose energy in the Ramachandran map is not drastically changed during the conversion like in β-sheets and γ-turns. In this case, they are likely to be stabilized by similar side-chain-to-side-chain interactions. This observation is consistent with results reported in literature [13,17,23,24]. As an example, Peggion and coworkers in 2009 [25] proposed a structure-function relationship study on a mimetic peptide of the Parathyroid hormone (PTH) spanning residues 1-11 (PTH(1-11)). This peptide is a ligand of the PTH type-I receptor and was studied through the synthesis and characterization of all-D PTH retro-inverso analogues. The retro-inverso RI-PTH(1-11) analogues showed a reduced biological activity compared to the parent peptide, because of the absence of the α-helical structure which could be induced by introducing an Aib residues on the N-terminal position [26]. 

For the design of RI peptides two aspects should be therefore considered: (i) The importance of the interactions of the backbone amide groups of the parent peptide with the receptor are important [12,13,27]. (ii) Maintenance of the original 3D structure in the retro-inverted peptide, that means retention of most hydrogen bonds formed intra-backbone and those between the backbone and the side chains. 

The current review focuses on the main applications of retro-inverso peptides as potential biotherapeutics with improved stability in vitro and in vivo. The interest around this fascinating subclass of molecules is driven by their potential use in a vast area of applications here reviewed, including diagnostics, cancer therapeutics, neurodegenerative diseases, and new antibiotics (as antimicrobial peptides). 

## 2. Anticancer Applications—Diagnostic

In the cure of cancer, side effects following conventional drug treatments are currently on the rise. A growing number of studies indicate that peptides, more specifically anticancer peptides (ACPs), could be new valuable options in this field. Peptides have the advantage of exhibiting reduced immunogenicity, excellent tissue penetrability, and low-cost manufacturability compared to bigger molecules like proteins and antibodies. Also, they are easily modified to improve the in vivo stability and the biological activity, leading to an increased utility and versatility for cancer therapy. For these reasons, an ever-growing number of anticancer peptides (ACPs) are being evaluated at various stages of clinical trials [28,29]. Cancer development is characterized by a variety of processes that include migration from the primary tumor site, invasion through the basement membrane, invasion of metastatic cells into blood vessels, and finally localization at second sites [30]. During the proliferation of cancer cells, the core of the tumor becomes deficient in nutrients and oxygen. Therefore, this kind of cancer cells have to up-regulate the expression of pro-angiogenic factors to stimulate new blood vessels into and around the tumor to allow it to grow up [31,32]. Angiogenesis, which is the formation of new capillaries from pre-existing blood vessels, is the driving event in this physiological process that also regulates embryogenesis, postnatal growth, reproductive function, and wound healing [33]. The other side of the coin are the pathological mechanisms of angiogenesis which play important roles in numerous human diseases, particularly in the growth and spread of cancers [33,34,35]. The suppression of pathological angiogenesis, which is principally driven by vascular endothelial growth factors (VEGF) and by their interaction with the receptors VEGFR1 and VEGFR2, set on the surface of endothelial cells [34], is indeed an efficient and clinically validated approach in cancer therapy. Beyond antibodies and soluble receptors, also several peptides have been designed and tested as inhibitors of the VEGF-VEGFR kinase axes in the tumor angiogenic cascade, with many that have been approved by the regulatory agencies [35,36] or have entered various clinical trials [37,38] to block tumor growth and angiogenesis [37]. 

In this field, Vicari et al., in 2011 [38] designed peptides able to mimic the VEGF-binding site on VEGFR-2, reproducing the loop formed by the antiparallel β-sheets β5 and β6. The linear synthetic peptide VEGF-P3(NC) was cyclized by oxidation of two cysteines enabling the formation of the twisted peptide VEGF-P3(CYC) underpinning an anti-parallel β-sheet structure VEGF-P3(CYC) (Table 1). The cyclic peptide VEGF-P3(CYC) showed an increased affinity for VEGFR-2 and an improved capability to inhibit VEGF-dependent signaling pathways compared with the parent linear peptides, highlighting the close relationship between the structure and the activity of the molecule. In this case, the transition from the L peptide to its retro-inverso analogue was not successful. Indeed, the RI analogue, named VEGF-RI-P4(CYC) was not able to block VEGF-VEGFR-2 interaction, although VEGF-RI-P4(CYC) and VEGF-P3(CYC) showed a similar global conformation in solution as demonstrated by Circular Dichroism (CD). These results indicated that the RI analogue, despite the similar side chains disposition, was unable to expose its binding key residues in positions favorable for interacting with the receptor. Upon a more detailed structural characterization, the loss of activity was related to a different arrangement of the backbone as compared to the parent peptide [39].

Using a combinatorial screening on VEGF-activated endothelial cells, the retro-inverso peptide _D_(LPR) was shown to target VEGFR1 and neuropilin-1 [40]. The motif _D_(LPR) was successful validated as strong inhibitor of retinal angiogenesis in retinopathy models when administered in an eye-drop formulation [41], showing the effectiveness of the approach on short peptides. 

**Table 1 ijms-22-08677-t001:** Names, peptide sequences reported in the review and their applications.

Name	Sequence ^1^	Application	Ref
**Anticancer Applications—Diagnostic**			
VEGF-P3(CYC)	I^76^TMQ^79^CG^92^IHQGQHPKIRMI^80^CE^93^MSF^96^ *	Inhibition angiogenesis	[38,39]
_D_(LPR)	_D_(Leu-Pro-Arg)	Inhibition retinal angiogenesis; Diagnostic	[41,42,43]
SP5	PRPSPKMGVSVS *	Drug delivery	[44,45]
uPAR_88–92_	SRSRY *	Maintaining chemotactic activity and triggers directed cell migration and angiogenesis	[46,47]
RI-3	Ac-_D_(Tyr- Arg-Aib-Arg)- NH_2_	Prevent extracellular invasion by tumor cells	[48]
_D_(RGD)	_D_(Asp-Gly-Arg)	Diagnostic	[49,50,51]
VS	SWFSRHRYSPFAVS *	Glioblastoma multiforme (GBM)	[52,53]
VAP	SNTRVAP *	Gliomas, glioma stem cells, vasculogenic mimicry and neovasculature	[54]
WSW	SYPGWSW *	Glioma cells and tumor neovasculature	[55]
BK	RPPGFSPFR *	Glioma cells	[56]
FP21	YTRDLVYGDPARPGIQGTGTF *	Ovarian cancer	[57,58,59]
T7	HAIYPRH *	Drug delivery	[60]
**Applications in Immunology**			
TG19320	(rty)_4_K_2_KG	IgG binding	[61,62]
VSVp	RGYVYQGL *	antigen surface of hepatitis B virus	[63]
OVAp	SIINFEKL *	antigen surface of hepatitis B virus	[63]
PS1	HQLDPAFGANSTNPD *	antigen surface of hepatitis B virus	[63]
HAI	HAIYPRH *	Crossing BBB	[64]
THR	THRPPMWSPVWP *	Crossing BBB	[64]
InsB:9–23	HLVEALYLVCGERGG *	Analogue of diabetogenic islet peptide—prevents T-cell activation in humanized model mice	[65,66]
**Application in Neurodegenerative Diseases**			
Amytrap	WKGEWTGR *	Blocking the oligomerization and aggregation of Aβ_1–42_	[67,68,69,70,71]
IAPP_11–20_	RLANFLVHSS *	Strong inhibitory effects on amylin aggregation in T2DM	[72]
β-syn_36–45_	GVLYVGSKT *	Reduction of amyloid fibril and oligomer formation	[73]
**Application in Antimicrobial Antibiotics**			
RI1018	rrwirvavilrv	Preventing formation of Biofilm	[74]
RI-JK6	rivwvrirrwqv	Preventing formation of Biofilm	[74]
RI-73	lwGvwrrvidwlr	Damaging the bacterial membrane	[75]
BMAP-28	GGLRSLGRKILRAWKKYGPIIVPIIRIG *	Broad antimicrobial activities	[76]

^1^ The sequences reported are those of the parent peptides * (L-residues), unless otherwise indicated, like reporting D residues as lower-case letters or adding a “D” before the sequence.

Another example of the use of RI analogues against the VEGF-VEGFR complexes was shown by Calvanese et al. [42] who designed a series of cyclic peptides embedding the retro-inverso (RI) version of the consensus sequences RPL/LPPR, corresponding to peptides capable of preventing VEGF binding to VEGFR1 [41] and VEGFR2 [77], and to specifically inhibit human endothelial cells (EC) proliferation in vitro [77]. Direct binding experiments of the peptides to VEGFR1 and VEGFR2 identified a peptide that bound both receptors with a K_D_ in the low micromolar range but with a significant selectivity for VEGFR1 respect to VEGFR2 thus showing a potential as VEGFR1-selective diagnostic probe.

The important properties of the of LPR peptide’s RI mimics were also recently demonstrated by Rezazadeh et al. [43]. Small size, high stability, high affinity for VEGF receptors and good distribution in tumor tissues were the characteristics of the _D_(LPR) peptide which was suggested as a good candidate for use as SPECT probe for molecular imaging of cancer. _D_(LPR) was indeed labeled with technetium-99m (^99m^Tc), which is the first-choice radionuclide in diagnostic nuclear medicine. The authors prepared two different _D_(LPR) analogues having sequences lprpK-HYNIC and HYNIC-Kplpr, both incorporating the HYNIC L-peptide (sequence L-His-L-Tyr-L-Asn-L-Ile-L-Cys) acting as a bifunctional chelating agent (BFCA) at the C- or N-termini of the targeting peptide. Their results demonstrated that _D_(LPR) could be labeled not only with diagnostic radioisotopes but also with therapeutic radioisotopes for both imaging and curative purposes. 

Another L-peptide, named SP5 (Table 1), that efficiently and specifically binds to the vasculature of tumors was discovered by in vivo phage display [44]. Its RI analogue, D-SP5 [45], was designed, prepared, and investigated and showed a stronger targeting ability to VEGF-stimulated HUVECs. Importantly, D-SP5 recognized the same binding site as the parent peptide, although the receptor was unknown. Authors also demonstrated that D-SP5 could be an effective agent for drug delivery for angiogenesis and a potential targeting vehicle for use in clinical cancer therapy. Also, D-SP5 was conjugated to micelles and loaded with doxorubicin (Dox), showing significantly stronger tumor inhibition efficiency compared to L-SP5 micelles/Dox, negative controls, and free Dox. 

Another attractive approach for the clinical management of metastases arising from solid tumors is the control of cell motility. One major regulator of cell migration frequently overexpressed and commonly targeted by therapeutics is the urinary plasminogen activator receptor (uPAR), also known as the urokinase receptor which promotes the chemotactic activity, the migration and the angiogenesis of cancer many cells [78]. The minimal fragment of uPAR spanning residues 88–92 [46] (Table 1) was identified as able to maintain the uPAR chemotactic activity and to trigger cell migration and angiogenesis properties in vitro and in vivo [47]. The activity of this peptide, known as uPAR_88-92_, was mediated by its direct interaction with the receptor FPR type 1 (FPR1), which is able to activate the vitronectin receptor [79]. Inhibition of the uPAR/FPR1 interaction represents an attractive target to inhibit the metastatic process in solid tumors. Carriero et al. [48] reported the tetrapeptide analogue RI-3, Ac-_D_(Tyr-Arg-Aib-Arg)-NH_2_ (Table 1), which at variance with the precursor peptide, was an antagonist of the uPAR/FPR1 interaction and as such could prevent in vivo extracellular matrix invasion, tumor cell infiltration into the blood, and capillary network formation. 

Integrins [80,81] are other receptors playing important roles in cell–cell and cell–matrix interactions during developmental and pathological processes [80,82,83]. They are highly correlated with angiogenesis, tumorigenesis, metastasis, and drug resistance [84,85,86]. In particular, integrin α_v_β_3_ isoform represents an interesting molecular target for many diagnostic and therapeutic applications [87,88,89,90]. α_v_β_3_ specifically recognizes the consensus tripeptide Arg-Gly-Asp (RGD), derived from extracellular matrix proteins, which has been largely exploited as radiolabeled carrier for the early diagnosis of malignant tumors [91]. Recently, Karimi et al. [49] presented a HYNIC-_D_(RGD) peptide labeled with the radioisotope ^99m^Tc. The radiochemical purity of HYNIC-_D_(RGD) was about 100%, showing the efficiency of this method to increase the quality of labeling compared to previous methods used for cyclic peptides [50]. At the same time, the retro-inverso portion of the peptide conferred higher stability in serum and higher affinity for integrins compared to the cyclic RGD parent peptide. Moreover, the peptide ^99m^Tc-HYNIC-_D_(RGD) showed radiochemical properties and in vitro targeting ability for human cancer cells similar to those reported in previous studies on other analogues [49]. Liu et al. [51] reported the in vitro investigation of several linear RI peptides based on the RGD motif conjugated to cell-penetrating peptides based on poly-arginines. Their results suggested that linear RI analogues were potentially useful as tumor targeting carriers with biological activity similar to RGD alone.

The peptide VS (Table 1) selected through the screening of a phage display library, also showed high binding affinity towards integrins, in particular against α_6_β_1_ and α_v_β_3_. The RI analogue [52] was designed, prepared and tested against glioblastoma multiforme (GBM), the most common and lethal tumor of the central nervous system [53]. Specifically, the RI variant of VS conjugated with PEG-PLA (poly-lactic acid) was used to prepare micelles which efficiently encapsulated doxorubicin (DOX), penetrated the tumor mass, and reduced its volume more efficiently compared to the control, the free drug, or other micelle formulations. These results showed that the Dox-loaded micelles functionalized with the RI-VS analogue had better anti-glioma effects in vivo, with fewer side effects compared with other formulations. 

The peptide VAP (Table 1) was shown to have high binding affinity in vitro to GRP78 protein, which is overexpressed in gliomas, glioma stem cells, vasculogenic mimicry, and neovasculature [54]. The prediction of binding for the analogue RI-VAP to GRP78 was similar to that of the parent peptide and, in addition, remarkable tumor accumulation was observed experimentally by imaging in vivo. RI-VAP-modified paclitaxel-loaded polymeric micelles had better anti-tumor efficacy compared to free taxol, to paclitaxel-loaded simple micelles, and to micelles modified with parent peptide. 

The short peptide WSW (Table 1) was reported to efficiently and selectively penetrate the blood–brain barrier (BBB) and blood–brain tumor barrier (BBTB) to reach glioma cells and tumor neovasculature, suggesting that it may be a suitable carrier for intracranial glioma targeting. Ran et al. [55] designed, synthesized and studied RI-WSW that exhibited higher endocytosis efficiency than the parent peptide. This property was explained by the higher targeting efficiency of the RI derivative and likely higher penetration efficiency. Moreover, micelles decorated on the surface with RI-WSW showed strong anti-angiogenesis and antitumor effects and increased penetration ability in vitro and in vivo toward tumor cells and angiogenic blood vessels. In a similar study Xie et al. [56] proposed a RI analogue of bradykinin, named RI-BK (Table 1), capable of crossing BBTB. The molecule was highly active and selective towards the bradykinin type 2 (B2) receptor, as also demonstrated by computational analyses. RI-BK was used to decorate paclitaxel (PTX)-loaded micelles whose accumulation was increased in glioma but not in normal brain. Co-administration of RI-BK increased the therapeutic efficiency of the drug-loaded nanocarriers in glioma. These results underscored the efficacy of glioma-targeted drug delivery, based on the use of micelles functionalized with retro-inverso peptides, in improving therapeutic efficacy for glioma treatment. 

Follicle-stimulating hormone receptor (FSHR) expression is limited to the reproductive system [92,93] and might be targeted to deliver drugs against ovarian cancer with high selectivity and specificity. In particular, nanoparticles carrying the RI variant of the peptide FP21 showed to bind FSHR (Table 1). They were thus used as an ovarian cancer targeted delivery system [57,58,59], showing improved biostability compared to the parent peptide, with no degradation even after 12 h incubation with proteolytic enzymes. The data obtained on the RI peptide encouraged further developments and optimizations of the molecule for treating ovarian cancers expressing FSHR [57]. 

In another study, Zhang et al. demonstrated that the RI derivative of the same peptide FP21 conjugated to nanocarriers had significantly enhanced anti-tumor effects working by reducing the tumor volumes in nude mice from 33.3% to 58.5%. This effect was likely amplified by the high resistance of the ligand to hydrolysis [57,58,59]. Another tumor promoter is the Transferrin receptor (TfR), an important transmembrane glycoprotein involved in iron transport. TfR is overexpressed in tumors because of the increased demand for iron during tumor rapid growing [94,95]. Recently, the RI derivative of the TfR-targeting T7 peptide (Table 1) was shown to have enhanced serum stability and higher binding affinity to TfR [60] than the parent peptide. This property was efficiently exploited modifying the surface of liposomes (LIP) to realize a tumor selective drug delivery system. The RI-T7-LIP particles exhibited significantly higher accumulation in tumors than T7-LIP and Transferrin-LIP. A complete pharmacokinetic study was performed to further investigate the potential of RI-T7-LIP in vivo using Docetaxel-loaded RI-T7-LIP which induced markedly increased apoptotic and necrotic areas in the treated mouse models.

## 3. Applications in Immunology

As previously observed, a strong topological correlation is at the base of antigenic cross-recognition between linear antigens and the corresponding retro-inverso isomers [96,97,98]. In several studies, monospecific murine antibodies were used as conformational probes to demonstrate the existence of surface similarities between a cyclic peptide mimicking the CD4 surface, which was a synthetic analogue of the third complementarity-determining region (CDR3) of immunoglobulins, and its corresponding retro-inverso isomer [60,96]. Anti-CD4 antibodies have been used to inhibit in vivo the clinical symptoms associated to the CD4-dependent auto-immune disorder allergic encephalomyelitis [22]. On this ground, RI analogues of CD4 loops were hypothesized as potential synthetic vaccines, as immunodiagnostics, and for the development of new generations of immunomodulators for the treatment of various CD4-related diseases [99]. For these reasons the cross-recognition of peptide surfaces by anti-CD4 antibodies was investigated using RI-peptide mimetics showing a strong correlation between antibody recognition and the simple arrays of side chains, with minimal contributions from the backbone atoms. These observations anticipated later studies showing that even simple arrays of alternating side chains, obtained by glycine-alternated peptide sequences (sequence-simplified peptide antigens), exposed on the same face of a peptide ideally adopting a fully extended conformation, were sufficient to retain the binding to antibodies. These insights were obtained studying the surfaces of sequence-simplified variants of retro-, inverso-, and RI derivative of parent 15-mer peptide antigens [100]. A series of polyclonal antibodies was generated in rabbits against 15-residue chimeric peptides and RI analogues able to bind interleukin 2 and to inhibit its interaction with the p55 interleukin 2 receptor subunit [101]. 

In diseases of inflammatory origin, such as systemic lupus erythematosus (SLE), one major event leading to a pathological condition is the interaction of immunoglobulins (IgG) with the corresponding cellular receptors. The pathogenic mechanism in SLE is the production of autoantibodies. To inhibit the interaction between IgGs and their receptor, a proteolytically stable form of the tripeptide Arg-Thr-Tyr tetramerized on a multilysine scaffold and obtained by inverting the chiral centers of the tripeptide’s amino acids (D-Arg-D-Thr-D-Tyr, TG19320) was used. The tripeptide bound the Fc portion of the antibodies [61], prevented the binding to Fc receptors and rescued from death transgenic mice harboring SLE-prone mutations [62].

Also, several negative examples of the use of RI peptides are reported in the current literature. Most parameters influencing the activity of these molecules have been described above. However, other possible reasons for their unsuccessful utilization are yet to be understood. Nair et al. carried out an interesting study to investigate the success or failure of retro-inverso isomers to mimic the corresponding all-L molecules in the case of antigenic epitopes (Table 1) [63]. They based their analysis on the T cell epitopes from vesicular stomatitis virus glycoprotein peptide (VSVp) and ovalbumin epitope (OVAp) and on the B cell epitope (PS1) derived from the antigen surface of hepatitis B virus. The parent VSVp and OVAp showed conformations similar to those of their corresponding RI analogues (both adopted extended conformations), and in the Ramachandran plots the distribution of ϕ and φ angles for the parent and the RI analogues occupied the same plot regions. On the contrary, in the case of the peptide PS1 and its RI derivative the two molecules showed distinct conformational propensities. Indeed, the parent peptide bound the antibody adopting a specific β-turn conformation that was not mimicked by the RI analogue. Although the plasticity of the epitope conformation in solution allowed a partial overlap of the angle values accessed by the RI analogue, the latter did not bind the anti-peptide antibody [63]. 

Recently, the RI analogues of a family of peptides capable to cross the blood–brain barrier (BBB), HAI and THR [64] were shown to possess improved protease-resistant properties and to maintain the original BBB shuttle activity of the parent peptide. However, the RI derivatives were much less immunogenic and as such provided an important improvement compared to the original molecules [102]. Another study reported the RI analogue of a peptide able to suppress T-cell activation in Type 1 diabetes mellitus (T1D) [103]. Currently, there are neither curative nor preventive treatments to block the auto-immune destruction of the islets of Langerhans (beta cells). A RI analogue of the diabetogenic islet peptide named InsB:9–23 (Table 1) [65,66], responsible of the auto-immune response, was shown to prevent T-cell activation in humanized model mice both ex vivo and in vivo. The peptide blocked the immune-mediated beta cells destruction, thereby suggesting a novel therapy for patients at earlier stages of T1D. The use of this molecule opened a highly positive clinical perspective since the treated animals showed a larger beta cells reserve compared to animals at later stages of the disease.

In this field, the use of bioinformatic platforms—now largely and freely accessible—is becoming an invaluable tool to quickly identify new peptides potentially suitable for developing synthetic vaccines and peptidomimetic therapeutics. Robson et al., for example, using bioinformatic tools identified the sequence KRSFIEDLLFNKV as a well conserved region around one of the known cleavage sites of the SARS coronavirus, used for cell entry by the virus itself. The authors proposed the use of a RI analogue and studied its conformational flexibility that might offer an advantage for the molecule’s action in vivo because of the capacity to better adapt on the target binding site. According with preliminary studies using molecular modeling and docking, the proposed RI peptide was expected to bind to the angiotensin converting enzyme type 2 (ACE2) which is the target of SARS-CoV in lung cells, and to work as an inhibitor able to prevent the proteolysis required for activation of the S spike protein [104].

## 4. Applications in Neurodegenerative Diseases

Deposition of protein fibrils is one of the leading causes of pathological conditions associated to neurodegenerative diseases. Protein fibrils form when a protein in β-pleated sheet conformation self-associates, mainly through hydrogen bonds, precipitating and generating protein deposits which accumulate in many different organs and tissues [105]. It is currently debated whether the precipitated insoluble fibrils or actually soluble oligomers are the cytotoxic aggregative elements working as diseases etiologic agents [106]. However, blocking or slowing down the aggregating phenomena is believed to be a major therapeutic option in this field. In the case of Alzheimer’s disease (AD), advanced approaches were initially based on direct immunization with Aβ_1–42_, although the first clinical trials were stopped due to adverse effects involving detrimental T-cell mediated brain inflammation [107]. Also, no significant improvements in terms of reduction of symptoms and immunological adverse reactions in patients [108] were observed with bapineuzumab, a humanized anti-Aβ_1–42_ monoclonal antibody (mAb). Very recently, aducanumab a mAb that binds only aggregated and soluble oligomers of Aβ has been approved for treating AD, strongly indicating that the anti-aggregation therapy with Aβ targeting molecules is an effective first line treatment for this disease. As alternative anti-aggregating agents, peptides capable to prevent Aβ_1–42_ aggregation or to dissociate preformed Aβ_1–42_ aggregates have been largely investigated. Recently, the synthesis of Amytrap, a tetrameric RI analogue of the all-L peptide WKGEWTGR has been reported. Amytrap was pegylated and conjugated to human serum albumin (HSA) to enhance its bioavailability [67] and as such displayed high affinity for the GSNKG region of Aβ_1–42_, blocking oligomerization and aggregation of the full length polypeptide. Using this molecule, the authors observed a significant reduction of Aβ_1–42_ levels in the brain as determined by immunohistochemical analyses of brain tissues. They also observed that Amytrap sequestered the soluble protein, shifting its ability to deposit into the brain. Previously, the same authors prepared the RI analogue of another peptide capable to block the oligomerization of Aβ_1–42_. The parent molecule was a chimeric peptide obtained by conjugating the HIV-1 “TAT” sequence to the Aβ fragment 16–20 (Aβ_16–20_). Its RI variant, named RI-OR2-TAT, was meant to act as a cell-permeable and brain-penetrant Aβ aggregation inhibitor [68]. With this molecule, the authors observed a rapid crossing of the BBB, an effective binding to the amyloid plaques and a reduction of the Aβ oligomers level of in the brain. Since RI-OR2-TAT inhibited Aβ aggregation at relatively high concentrations [68], recently it was covalently attached to nanoliposomes (NLs) using the ‘click’ chemistry. An efficient crossing of the BBB in in vitro models was observed and lower concentrations of this form of the peptide were enough to inhibit aggregation of Aβ. Also, protective effects towards the toxicity exerted by pre-aggregated Aβ on neuronal cells were observed in vivo, preventing memory loss in transgenic mice. The presence of NL improved the potency of RI-OR2-TAT due to the multivalent effect deriving from the presence of multiple copies of peptides decorating each liposome [69]. Morris et al. [70] reported a pre-clinical study of the same molecule labeled with ^18^F, [^18^F]RI-OR2-TAT, to demonstrate its in vivo stability and the hepatobiliary route as the primary excretion pathway of the intact peptide. These results were the base of a study where RI-OR2 was modified by replacing hydrophobic amino acids with non-natural building blocks. The final peptidomimetic, even more resistant to proteolytic degradation, retarded the aggregation of Aβ_1–42_ and also partially dissolved newly aggregated oligomers [71]. 

Another approach proposed to combat Alzheimer’s disease is blocking the initial cleavage of the amyloid-β protein precursor (AβPP) by the β-site AβPP cleaving enzyme 1 (BACE1). Some RI-analogues based on a fragment of AβPP were synthesized as chimeras with the TAT carrier to facilitate cell membrane permeation and crossing of the BBB. The authors observed a decrease of both Aβ_1–40_ and Aβ_1–42_ (Aβ_1–40/42_) production without inducing cytotoxicity. Moreover, Aβ_1–40/42_ levels decreased in plasma and brain, diminishing also the levels of soluble AβPP production and of insoluble Aβ following chronic treatments. These results suggested a possible use of the chimeric RI peptides as a selective disease-modifying therapy for AD [109].

A further application of retro-inverso peptides is to prepare nanocarriers and delivery systems as new tools in Alzheimer’s disease. A recent approach was based on silencing BACE1 using RNA interference (RNAi). In particular using small interfering RNAs (siRNAs) some authors presented a “dual targeting” strategy based on nanoparticles (NP) built with a peptide component and a modified polyethylene glycol [110]. The peptide moiety was a BBB targeting peptide [111], more specifically it was the RI analogue of the all-L peptide TGNYKALHPHNG. The resulting NPs showed low toxicity and high transfection efficiency and were able to transfer siRNAs in the brain [112]. Using this system, the authors observed an increase of the BBB-penetration, of the neuron-targeting efficacy and higher neuroprotective effects reflected by improved cognitive performance. Also, the downregulation of the protein-tau phosphorylation level, the promotion of the axonal transport and the attenuation of microgliosis were observed in mice model [111] following treatment with these NPs. 

Recently, molecular dynamics was used to study the formation of fibrils between the RI-Aβ_1–40/42_ and the parent Aβ_1–40/42_ to elucidate the mechanism of cross-fibril formation and the effect of RI-Aβ_1–40/42_ on fibril stability. The resulting models indicated that Aβ_1–40/42_ and RI-Aβ_1–40/42_ generated a two-layer structure with similar stability. In particular, the dihedral angles were of opposite sign for the Aβ_1–40_ fibrils and the extent of the twists was different. Furthermore, the twists of RI-Aβ_1–42_ and of the parent peptide were close to zero. Analyzing the RI-Aβ fibrils, the authors observed that the number of hydrogen bonds connecting the chains within the fibril was lower compared to the parent peptide fibrils. The average number of missing hydrogen bonds was 7, mostly in the region around residues 23–29, and this strongly impacted on the different stability observed between the two fibrils. Data also suggested that the full-length RI-peptides could support fibril formation and their presence led to a decreased amount of soluble toxic Aβ oligomers [10]. These observations were in agreement with the experimental observations reviewed in this work.

In addition to AD, also type 2 diabetes mellitus (T2DM) and Parkinson’s disease (PD) are related to amyloidogenesis. In T2DM, the Islet amyloid polypeptide (IAPP also known as amylin) aggregates into β-pleated sheet structures damaging pancreatic islet β-cells. The “hot spot” peptide segment encompassing residues 8–18 (IAPP_8–18_, sequence ATQRLANFLVH) represents the “sticky” region of human IAPP, which is also able to assemble with IAPP_22–28_, sequence NFGAIL [113,114]. In order to inhibit the early stages of IAPP hetero- and self-aggregation, a library of RI peptides covering the region 11–20 of IAPP (Table 1), was generated and studied evaluating their impact on the fibrillogenesis properties of full-length human IAPP [72]. The authors found a RI non-toxic analogue showing strong inhibitory effects on amylin aggregation, as confirmed by negative stain electron microscopy (TEM). Inquisitively, the RI-analogue alone aggregated already at low concentrations. The authors also introduced N-methylation as a way to prevent H-bond formation and avoid aggregation [115]. The new N-methylated RI variant showed a clear dose-dependent inhibition of fibril formation and was stable against an ample range of different proteolytic enzymes and in human plasma.

Shaltiel-Karyo and colleagues studied the inhibition of oligomerization of α-synuclein (α-syn) [116], a protein whose structural deformation is associated with PD. The isoform β-synuclein (β-syn) is a natural inhibitor of the aggregation of α-syn [73]. The entire sequence of β-syn was then systematically mapped using synthetic analogues to identify the domains able to mediate the molecular recognition between β-syn and α-syn. A synthetic RI-analogue of the 36–45 β-syn fragment (sequence GVLYVGSKTR) was able to reduce both amyloid fibril and soluble oligomer formation in vitro. The authors also tested the RI-analogue in a *Drosophila* model expressing a mutated α-syn in the nervous system and observed a reduction of α-syn accumulation in the brains of the flies, thus suggesting that this approach can pave the way for developing a novel class of therapeutic agents to treat PD in the future.

## 5. Application of RI Peptides as Antimicrobial Antibiotics

The systematic and widespread misuse and abuse of antibiotics has made antibiotic resistance a major medical complication following hospitalization [117,118]. The World Health Organization has identified a list of “priority pathogens”, both Gram-positive and Gram-negative, which represent the biggest threat to human health caused by multidrug-resistant bacteria [119]. Among these microorganisms, those collected under the acronym “ESKAPE” (i.e., *Enterococcus faecium*, *Staphylococcus aureus*, *Klebsiella pneumoniae*, *Acinetobacter baumannii*, *Pseudomonas aeruginosa*, *Enterobacter* spp.) are those needing the urgent and prompt discovery of new antimicrobials. Many surgical procedures, or medical treatments that suppress immune system will become impracticable due to infections by antimicrobial resistant pathogens. Also, prophylactic treatments that are normally effective become inefficient. Today, the containment of infections by these microorganisms is very problematic [120] and new antibiotic drugs are urgently needed. An alternative to conventional antibiotics would be the use of antimicrobial peptides (AMPs) [121]. They are widely spread in nature being present in bacteria as well as in higher eukaryotes and play an important role in innate immunity and in both adaptive and non-adaptive immune responses [122]. Their antimicrobial action is based on multiple mechanisms that together contribute to eliminate pathogens [123,124]. However, despite the interesting and very promising antimicrobial effects displayed by many AMPs, so far only 10 peptide-based antimicrobials have reached the clinical use [120]. 

Indeed, several peptides have shown nephrotoxic or hemolytic side effects, strongly discouraging their use as drugs. Given their toxicity, the use of some approved AMPs, such as colistin, is relegated among the last treatment options against multi drug-resistant Gram-negative infections [125]. Nevertheless, novel AMPs can be obtained choosing among a vast repertoire of sequences and structures and novel peptides candidate as potential therapeutics are continuously developed at least at preclinical level. Starting from naturally occurring AMPs, synthetic derivatives have been rationally designed in order to maintain the antimicrobial pharmacophores, to improve the resistance to proteolysis, to reduce cytotoxicity and possibly improving the activity [126,127,128].

A recent application of the retro-inverso approach to antimicrobial peptides has been reported by Neubauer and colleagues, although this procedure not always results in enhanced antimicrobial activity [129]. The antimicrobial and hemolytic activities of a set of 6 AMPs were investigated together with their hydrophobicity, secondary structure content, and ability to self-associate. The antimicrobial peptides were aurein 1.2, CAMEL, citropin 1.1, omiganan, pexiganan, and temporin A together with their retro-inverso analogues. These peptides were selected for their broad-spectrum activity against fungi, bacteria and also for their possible anticancer applications (Table 2). Of interest, CAMEL, omiganan, pexiganan, and temporin A are in clinical trials, with potential uses against some ESKAPE bacteria strains (Table 2 and Table 3). Of the compounds studied, the majority displayed antimicrobial activity (Table 3), although in most cases there was a decrease of the antibacterial potential with respect of the native molecule. In fact, only the RI omiganan displayed enhanced antimicrobial activity mainly against Gram-negative bacteria compared to parent peptide. Similarly, retro-inverso pexiganan exhibited a good activity towards *K. pneumoniae* and *P. aeruginosa.*

In the list reported in Table 3, CAMEL is the only chimeric peptide, designed by Merrifield in 1995 [146], containing fragments of two peptides with different antimicrobial activities [147]. CAMEL, which is in preclinical trial, was one of the strongest antimicrobial peptides, but its RI analogue was only active toward *K. pneumoniae* and *P. aeruginosa*. Interestingly, the secondary structure of the peptide was not the prerequisite for establishing significant interactions between the peptide and its biological target and the antimicrobial activity was only due random interactions with the core lipidic membrane of the pathogen [146]. The differences in antimicrobial activity between the peptide and the RI analogue could therefore not be explained. 

A major cause of antibiotic resistance is the formation of biofilms, which arise from bacteria growing on surfaces or at the air-liquid interfaces as a response to exogenous stresses. In biofilms, bacteria are encased in a protective extracellular matrix containing water, polysaccharides, proteins, extracellular DNA, and lipids [148]. de la Fuente-Núñez and colleagues reported the synthesis and analysis of a library of peptides and their RI-analogues to eradicate biofilms produced by *Pseudomonas aeruginosa* [149]. They observed that the RI-analogues (Table 1) named RI1018 and RI-JK6 were more potent at stimulating degradation and/or preventing accumulation of the stress-related second messenger nucleotide guanosine penta- and tetra-phosphate [(p)ppGpp] which plays an important role in biofilm development in many bacterial species [74]. They also demonstrated that these analogues killed bacteria growing as biofilms, which have a high adaptive resistance and are difficult to eradicate. Moreover, these peptides had synergic effect with common antibiotics, rendering biofilms more susceptible to their attack. Another AMP able to damage the bacterial membrane is a truncated and modified RI-analogue of Aurein 2.2 (RI-73, Table 1), which was recently used to eradicate preformed *Staphylococcus aureus* biofilms [75]. The antimicrobial activity of these analogues was increased 2- to 8-folds and when conjugated with biocompatible polyethylene glycol (PEG)-modified phospholipid micelles their toxicity toward human cells and aggregation were strongly reduced. Although RI-73 exhibited a good activity, the PEG-conjugated analogue showed a partially reduced activity. 

A further public health problem in many countries throughout the world is represented by the insurgence of multi drug-resistance against protozoan parasites, such as *Leishmania*. Host defense peptides (HDPs) are becoming promising options for new therapies. HDPs have the advantage of their small size and their amphipathic and cationic character that is able to induce permeabilization of cell membranes. Cathelicidins, a family of HDPs, have shown significant antimicrobial activities against various parasites including *Leishmania* spp. [150]. In particular, a study was carried out using the bovine myeloid antimicrobial peptide 28 (BMAP-28, Table 1), a cathelicidin with broad antimicrobial activities, and its inversed and RI-analogues [76]. The study demonstrated that D- and RI-BMAP-28 were also effective antimicrobials against *Leishmania major*, working in a dose dependent manner with a mechanism leading to disruption of membrane integrity [151]. Thus, the protection conferred by RI-BMAP28, accompanied by a reduced toxicity and increased stability, could be exploited to develop effective antimicrobial therapeutics [152].

## 6. Conclusions and Future Perspectives

In the field of peptidomimetics, retro-inversion has been largely explored to improve peptide stability while retaining the parent molecule’s activity. Changing the order of the amino acids and their configuration has been also a mean of introducing novelty and to overcome existing intellectual property claims [153]. The first examples of their use were reported by M. Goodman in the mid-1970s [154], who was interested in the study of stereochemical and conformational properties of retro–inverso (RI) amide bonds in linear peptides. Interesting examples were next reported by Merrifield with studies on the CAMEL peptide [146,147], which was a chimeric peptide derived from the merging of two AMP. Despite the amazing results reported in literature, the application of retro-inversion to generate peptidomimetics is still rare or however uncommon. 

As also evidenced in this review, several studies have indeed reported that the general and straightforward process of retro-inversion becomes more likely effective with very short sequences where conformation plays a limited or no role and activity is mostly due to a simple array of side chains. For instance, Sakurai’s results [155] suggested that the interaction between the RI analogue of VWRLLAPPFSNRLL and the ganglioside GM1, a glycolipid with high affinity for the cholera toxin subunit B (CTB), was mediated only by the peptide side chains while those of the backbone, whose direction was thus irrelevant, were completely negligible. One could thus expect that a RI analogue can better mimic the parent peptide when the free energy of interaction of the backbone with all other atoms is insignificant for the stability of the peptide 3D structure.

Beyond these basic rules applicable to short peptides or other specific examples, the reasons for the frequent failure of RI isomerization of longer molecules are still largely unclear, and definite instructions for possibly improving the success rate are unresolved. The reversal of the peptide backbone and the shift of the H-bond network it is involved into is a major alteration of the fine equilibrium of the forces that supports the conformation of a peptide having an organized 3D structure. Therefore, as for the folding of a natural molecule, the lack of one such important puzzle piece prevents the correct assembling of the structure although the side chains may potentially have access to the same conformational space of the parent molecule. We can thus conclude that the design of a successful retro-inverso analogue of a folded peptide has the same complications as for the *de novo* design of a new protein or peptide and one should thus proceed following the rules, still not well understood and codified, of protein folding, exploiting and using the geometrical and structural features of amino acids in D configuration. For example, the RI isomerization and structure reconstruction of the MDM2/MDMX peptide inhibitor stingin, which adopts an N-terminal loop and a C-terminal α-helix, lead to an isomer that partially retained binding (3.0–3.4 kcal/mol reduction) and showed a decreased ability to prevent the interaction with p53 [11]. These conformation and energy issues have been often discouraging because of the frequent loss of biological activity observed in larger molecules showing well-defined tridimensional organizations. Merrifield indeed soon observed that the efficiency of peptide retro-inversion was not only related to inversion of its chirality but to the global change of the 3D conformation [146]. These observations have been indirectly confirmed showing that retro-inverso analogues of unstructured peptides more often maintain or even increase the activity compared to the parent peptide [12]. 

On the other hands, peptides that assemble into β-sheets adopting extended conformations establish a large and well-organized network of interactions, mostly H-bonds, with the adjacent molecules. Also, the side chains are well packed each other. In this case, despite the strong backbone interactions, retro-inverso analogues have more chance to be successful if the registry of H-bonds and of side chain-to-side chain interactions is corrected to account for the inverted amide bonds. The molecular dynamic simulations of amyloid fibrils in AD [10] or amylin in T2D [156] indeed showed that the interactions of both side chains and backbone of RI peptides were re-aligned establishing different patterns of contacts and hydrogen bonding. Also, the twist of the RI analogue β-sheets was similar and the complex had only slightly lower stability compared to the parent peptides.

Computational approaches might be of great help and might open a new season in this field as suggested by Robson [104]. Despite their many limitations, we believe their use still has a place in the design of drugs based on bioactive peptides. This belief stems from the simplicity of the design, from the rapidity in making synthetic peptides and from the immediate benefits resulting when the molecules maintain their activity. Therefore, this review would be an incentive to continue working with these types of molecules, also to further investigate the conformational and topological space they need to occupy to fully mimic bioactive peptides with complex structure.

## Figures and Tables

**Figure 1 ijms-22-08677-f001:**
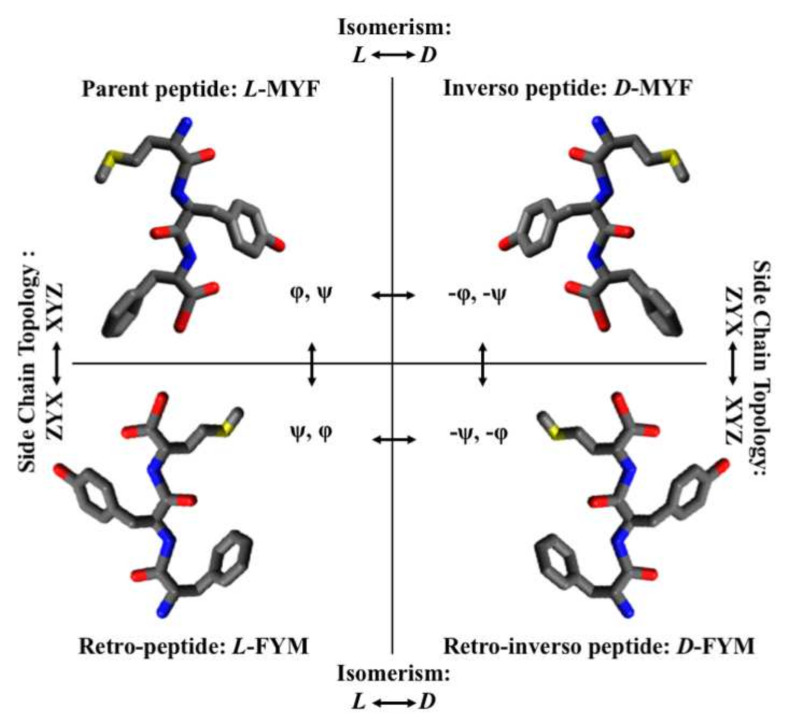
Topological relationship between a peptide and its inverso, retro, and retro-inverso analogues, illustrated for the example peptide MYF [17]. As shown, the topology of the side chains of the retro-inverso analogue, in the C-to-N orientation, is the same as the parent peptide in the N-to-C orientation (figure adapted by [13]).

**Table 2 ijms-22-08677-t002:** Peptide-based antimicrobial compounds in clinical trials and their retro-inverso analogues, based on [118,129].

Peptide	Sequence	Net Charge	Helicity ^a^		% ACN ^b^	Application	Mechanism of Action	Status	Therapeutic Indication	Ref.
			SDS	DPC						
aurein 1.2	GLFDIIKKIAESF-NH_2_ *	+1	<	>	43.93	Antimicrobial and anticancer properties	Prerequisite aggregation and carpet-like mechanism	in vitro [129]		[130]
RI-aurein 1.2	fseaikkiidflg-NH_2_ **		>	=	37.10					
CAMEL	KWKLFKKIGAVLKVL-NH_2_ *	+6	=	<	33.73	Broad spectrum antibacterial	Bacterial membrane disruption	Preclinical [119]	Bacterial infections [119]	[131,132]
RI-CAMEL	lvklvagikkflkwK-NH_2_ **		>	=	30.71					
citropin 1.1	GLFDVIKKVASVIGGL-NH_2_ *	+2	=	=	42.40	Broad spectrum antibacterial and anticancer properties	Prerequisite aggregation and carpet-like mechanism	in vitro [129]		[133]
RI-citropin 1.1	lggivsavkkivdflg-NH_2_ **		=	>	41.28					
Omiganan	ILRWPWWPWRRK-NH_2_ *	+5	NO	NO	32.92	Broad spectrum antifungal, antibacterial	Bacterial membrane disruption	Phase III complete (discontinued) Phase III complete Phase III on going Phase II complete Phase II complete Phase II complete Phase II complete Phase II on going [118]	Local catheter site infections Topical skin antisepsis Papulopustular rosacea Acne vulgaris Atopic dermatitis Vulvar intraepithelial neoplasia Condylomata acuminata (external genital warts) Facial seborrhoeic dermatitis [118]	[134,135,136]
RI-omiganan	krrwpwwpwrli-NH_2_ **		NO	NO	35.48					
Pexiganan	GIGKFLKKAKKFGKAFVKILKK-NH_2_ *	+10	=	=	30.58	Broad spectrum antibacterial	Bacterial membrane disruption	Phase III complete; rejected, efficacy not superior to current therapies [118]	Infected diabetic foot ulcers [118]	[137,138]
RI-pexiganan	kklikvfakgfkkakklfkgig-NH_2_ **		<	<	26.36					
temporin A	FLPLIGRVLSGIL-NH_2_ *	+2	<	>	42.80	Gram-positive bacteria	Bacterial membrane disruption	Preclinical [119]	Bacterial infections [119]	[139,140]
RI-temporin A	ligslvrgilplf-NH2 **		<	<	38.91					

* All-L sequences are reported as capital letters. ** Lower case letters indicate amino acids in the D configuration. **Note**: ^**a**^: The symbol =; >; < is referred to the helicity fraction calculated as in [141]. In particular, = means around 50%, > and < more or less 50%, respectively. Experimental conditions: CD spectra of the peptides were acquired in 10 mM phosphate buffer pH 7.4, containing SDS (sodium dodecyl sulfate) and DPC (dodecylphosphocholine) using a Jasco J-815 spectropolarimeter. All measurements were conducted using 0.15 mg/mL peptide solutions at 298 K [142]. ^**b**^: Hydrophobicity was determined by HPLC and was expressed as the % *v*/*v* acetonitrile at the retention time of the peptides (tR) [129].

**Table 3 ijms-22-08677-t003:** MIC values (μg/mL) of anti-microbial peptides and of their retro-inverso analogues against reference strains of microorganisms [143]. Taken from reference [129].

	*Gram-Positive*			*Gram-Negative*		
Peptide	*E. faecalis* ^1^ PCM 2673	*S. aureus* ^1^ ATCC 25923	*S. pneumoniae* ATCC 49619	*E. coli* ATCC 25922	*K. pneumoniae* ^1^ *ATCC 700603*	*P. aeruginosa* ^1^ *ATCC 9027*
Aurein 1.2	64	128	64	128	16	256
RI-aurein 1.2	256	>256	256	256	128	>256
CAMEL	8	4	0.5	2	0.125	2
RI-CAMEL	64	128	128	128	2	8
citropin 1.1	32	16	32	32	16	128
RI-citropin 1.1	128	64	128	64	32	>256
Omiganan	16	16	8	16	8	16
RI-omiganan	16	8	8	8	4	4
Pexiganan	16	8	1	4	1	2
RI-pexiganan	64	128	4	8	0.125	2
Temporin A	64	4	>256	256	128	>256
RI-temporin A	256	64	>256	256	128	256

^1^ These bacterial pathogens are comprised in the acronym ESKAPE, which are a group of Gram-positive and Gram-negative bacteria able to evade commonly used antibiotics due to their ever increasing multi-drug resistance (MDR) [117]. They represent the major cause of life-threatening nosocomial infections in immunocompromised and critically ill patients [144]. The acronym ESKAPE is based on the scientific names of six bacteria, *Enterococcus faecium*, *Staphylococcus aureus*, *Klebsiella pneumoniae*, *Acinetobacter baumannii*, *Pseudomonas aeruginosa*, and *Enterobacter* spp. In particular, *P. aeruginosa* and *S. aureus* are some of the most ubiquitous pathogens found in highly resistant biofilms [126,145].

## Data Availability

No datasets have been used for this study.

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
