# Peer review of "Recent Applications of Retro-Inverso Peptides"

_ijms, 2021, doi:10.3390/ijms22168677_

Round 1
Reviewer 1 Report
Dear Authors
i have no particular commnets on this review, being a collection of retro-inverso peptides and their main application in therapy.
The review is interesting in specialyzed field of peptide chemistry, that's why I found it more suitable for a peptide-oriented journal.
However it could be of some interest for everyone and it is easy to read.
My suggestion is minor after checking the english for some typos and grammar mistakes.
Author Response
Q1: i have no particular commnets on this review, being a collection of retro-inverso peptides and their main application in therapy.
The review is interesting in specialyzed field of peptide chemistry, that's why I found it more suitable for a peptide-oriented journal.
However it could be of some interest for everyone and it is easy to read.
My suggestion is minor after checking the english for some typos and grammar mistakes.
R1: We thank the reviewer for the appreciation of our work. We have thoroughly revised the English in the new version of the manuscript.Reviewer 2 Report
The authors have generated an interesting review of the literature on retro-inverso peptides. I have a few comments.
The most significant suggestion for the authors would be to try and include a more generalizing paragraph on the principles that may be deducted from the studies they reviewed. Although quite a number of applications and experimental approaches have been included and discussed, I miss a more aggregated view on the field: what is the essence of the many studies, which types of peptides, which types of peptide receptors could be named that are particularly suitable to be targeted by retro-inverso peptides? The general notion of RI analogues potentially being useful with unstructured peptides and those with a low contribution of the backbone is well taken, but are there other principles that may guide researchers when considering RI analogs? As stated above, this should most likely be summarized in a special paragraph.
p1, line 31: degraded, resulting in
p1, line 35: approaches
p2, line 66: Therefore, this kind of
p15, line 595: their configuration has also been a means of introducing
p15, line 596: property claims
Author Response
Reviewer #2
The authors have generated an interesting review of the literature on retro-inverso peptides. I have a few comments.
R1: We thank the reviewers for appreciating our work.
Q1: The most significant suggestion for the authors would be to try and include a more generalizing paragraph on the principles that may be deducted from the studies they reviewed. Although quite a number of applications and experimental approaches have been included and discussed, I miss a more aggregated view on the field: what is the essence of the many studies, which types of peptides, which types of peptide receptors could be named that are particularly suitable to be targeted by retro-inverso peptides? The general notion of RI analogues potentially being useful with unstructured peptides and those with a low contribution of the backbone is well taken, but are there other principles that may guide researchers when considering RI analogs? As stated above, this should most likely be summarized in a special paragraph.
Q2: We really thank the reviewer for his/her suggestion and we agree with him/her regarding the lack of a more generalized and aggregated compendium of the principles that might be applied in designing retro-inverso analogues of bioactive peptides. However, no matter how hard we try, general principles beyond those indicated and inherent in short or unstructured peptides are difficult to identify. Our suggestion is that, with larger molecules adopting well-organized secondary structures, one should proceed with a de novo design and following the rules, still not well understood and codified, of protein folding, exploiting and using the geometrical and structural features of amino acids in D configuration.
We have included this concept in the revised paragraph of Conclusions (now renamed Conclusions and Future Perspectives) which has also been largely implemented with a reflection on peptides forming aggregated beta-sheet structures. In this case the molecules are all in extended conformation and although they are held together by a dense network of hydrogen bonds and side-chain interactions, it would be possible in principle to design retro-inverted analogues with the same features as the precursors but paying attention to re-align and restore the hydrogen bond register in the inverted backbones.
Q3: p1, line 31: degraded, resulting in
p1, line 35: approaches
p2, line 66: Therefore, this kind of
p15, line 595: their configuration has also been a means of introducing
p15, line 596: property claims
R2: The suggested changes have been introduced together with a deep revision of the English language. Changes are visible with the text tracking option.